EMBO
Molecular Medicine

# Genomic structural variations lead to dysregulation of important coding and non-coding RNA species in dilated cardiomyopathy

Jan Haas[1,2,†], Stefan Mester[1,2,†], Alan Lai[1,2], Karen S Frese[1,2], Farbod Sedaghat-Hamedani[1,2], Elham Kayvanpour[1,2], Tobias Rausch[3], Rouven Nietsch[1], Jes-Niels Boeckel[1,2], Avisha Carstensen[1], Mirko Völkers[1,2], Carsten Dietrich[4], Dietmar Pils[5,6], Ali Amr[1], Daniel B Holzer[1], Diana Martins Bordalo[1,2], Daniel Oehler[1,2], Tanja Weis[1,2], Derliz Mereles[1,2], Sebastian Buss[1], Eva Riechert[1,2], Emil Wirsz[4], Maximilian Wuerstle[4], Jan O Korbel[3] , Andreas Keller[7] , Hugo A Katus[1,2], Andreas E Posch[4,**] & Benjamin Meder[1,2,*] 

## Abstract

The transcriptome needs to be tightly regulated by mechanisms that include transcription factors, enhancers, and repressors as well as non-coding RNAs. Besides this dynamic regulation, a large part of phenotypic variability of eukaryotes is expressed through changes in gene transcription caused by genetic variation. In this study, we evaluate genome-wide structural genomic variants (SVs) and their association with gene expression in the human heart. We detected 3,898 individual SVs affecting all classes of gene transcripts (e.g., mRNA, miRNA, lncRNA) and regulatory genomic regions (e.g., enhancer or TFBS). In a cohort of patients ($n = 50$) with dilated cardiomyopathy (DCM), 80,635 non-protein-coding elements of the genome are deleted or duplicated by SVs, containing 3,758 long non-coding RNAs and 1,756 protein-coding transcripts. 65.3% of the SV-eQTLs do not harbor a significant SNV-eQTL, and for the regions with both classes of association, we find similar effect sizes. In case of deleted protein-coding exons, we find downregulation of the associated transcripts, duplication events, however, do not show significant changes over all events. In summary, we are first to describe the genomic variability associated with SVs in heart failure due to DCM and dissect their impact on the transcriptome. Overall, SVs explain up to 7.5% of the variation of cardiac gene expression, underlining the importance to study human myocardial gene expression in the context of the individual genome. This has immediate implications for studies on basic mechanisms of cardiac maladaptation, biomarkers, and (gene) therapeutic studies alike.

**Keywords** cardiac transcriptome; dilated cardiomyopathy; expression quantitative trait locus; genomic structural variation; heart failure
**Subject Categories** Cardiovascular System; Chromatin, Epigenetics, Genomics & Functional Genomics

## Introduction

The myocardium has to permanently adapt to changes in the hemodynamic demand (Heusch *et al*, 2014), aging of the organism (Boon *et al*, 2013), and multiple external stressors (Ware *et al*, 2016). The coordination of these cascades involving cardiac energy metabolism, calcium handling, contractile elements, or protein turnover is not well understood, but ultimately executed by changes in gene transcription or protein translation (Correll *et al*, 2015; Nickel *et al*, 2015; Anderson *et al*, 2016; Mizushima *et al*, 2016; Tuomainen & Tavi, 2017).

Diverse mechanisms are known to contribute to adaptive and maladaptive gene expression in the human myocardium, such as transcription factors and their binding sites, micro- and circular RNAs, lncRNAs, histone modifications, and direct chemical changes of the DNA (Haas *et al*, 2013; Chang & Han, 2016; Lighthouse & Small, 2016; Zhao *et al*, 2017). Opposing to the dynamic and (mal)adaptive nature of these mechanisms are static effects on the

1   Department of Internal Medicine III, University of Heidelberg, Heidelberg, Germany
2   DZHK (German Centre for Cardiovascular Research), Heidelberg, Germany
3   EMBL (European Molecular Biology Laboratory), Heidelberg, Germany
4   Strategy and Innovation, Siemens Healthcare GmbH, Erlangen, Germany
5   Siemens AG, Corporate Technology, Vienna, Austria
6   Section for Clinical Biometrics, Center for Medical Statistics, Informatics, and Intelligent Systems (CeMSIIS), Medical University of Vienna, Vienna, Austria
7   Department of Bioinformatics, University of Saarland, Saarbrücken, Germany
    *Corresponding author. Tel: +49 6221 5639564; Fax: +49 6221 564645; E-mail: benjamin.meder@med.uni-heidelberg.de
    **Corresponding author. E-mail: andreas.posch@curetis.com
    †These authors contributed equally to this work

transcriptome originating from genetic variation. While yet little is known about the impact of these variations on the cardiac transcriptome and consequently phenotype, first pilot studies could link single nucleotide polymorphisms (SNPs) and cardiac gene expression (Koopmann *et al*, 2014).

Recently, analysis of the 1000 Genomes Project revealed a surprisingly large number of structural genomic variations in the human genome. SVs include large deletions, duplications, inversions, and complex rearrangements of stretches of DNA. In total, over 68,000 SVs were identified in the germline of population-based control subjects. Their cumulative size renders them to be the by far largest cause of genetic variability in humans (Genomes Project Consortium *et al*, 2015; Sudmant *et al*, 2015). Accordingly, it is estimated that four to five times more DNA letters are changed due to SVs compared to usually studied single nucleotide variants (SNVs) and a recent study in human tissues of deceased individuals linked about 8% of heritable gene expression variation to this under-investigated class of genome variation (Chiang *et al*, 2017). The impact of such changes on the transcriptome in the heart of patients is unknown.

In the present study, we used a multi-omics design to study the presence of SVs in a cohort of patients with heart failure due to dilated cardiomyopathy (DCM) and linked the genomic aberrations to myocardial gene expression by performing heart-specific SV-eQTL and SV-load correlations. By comparing the results to the regulation in blood of the same patients and stringent validation of the SV events including array-, PCR-based, and nanopore sequencing approaches, we provide a unique overview of a novel class of transcriptional regulators in the heart.

## Results

### Study population for multi-omics analysis

For this multi-omics analysis, it was required that high-quality material and biopsies were present of each DCM patient (Fig 1A). Included patients with suspicion for primary DCM underwent extensive clinical phenotyping including coronary angiography with

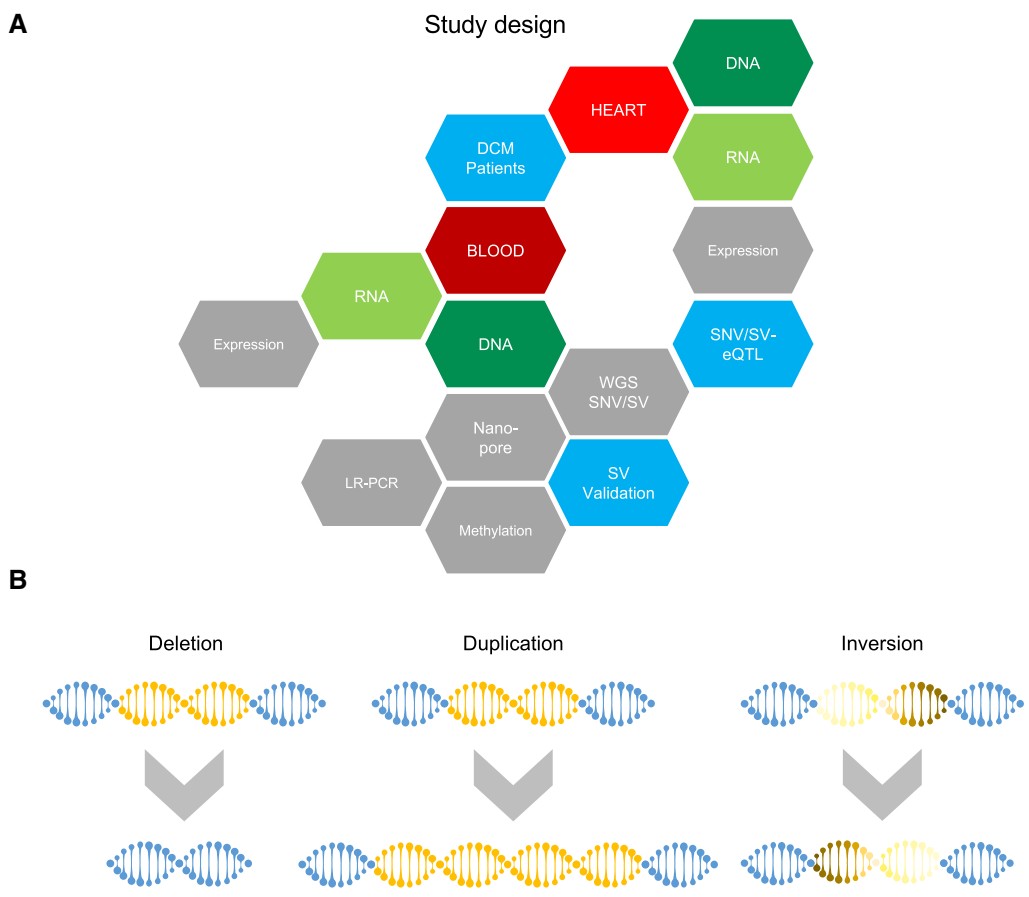

**Figure 1.  Study design.**

A    DNA and total RNA of DCM patients were isolated from left ventricular myocardial biopsies and peripheral blood. Next-generation sequencing was performed to obtain whole-genome sequences (WGS) and genome-wide transcription profiles. In addition, methylation profiles were assessed using Illumina 450K chip assay and used for validation of genomic structural variants (SVs). Long-range PCR (LR-PCR) and long-read nanopore sequencing were performed to exemplarily validate SVs. Identified SVs and SNVs from WGS were correlated to expression quantitative traits in an SNV/SV-expression Quantitative Trait Loci analysis.

B    Scheme visualizing the different structural variations studied. In a large deletion, distinct genomic loci are completely deleted on a genomic strand, whereas a duplication leads to multiple copies of a genomic loci. Inversions contain genomic sequences in an opposite direction.

**Table 1.  Baseline characteristics of analyzed patients.**

| Basic characteristics | (*n* = 50) |
|---|---|
| Age, mean ± SD, years | 53.8 ± 12.6 |
| Age at onset ± SD, years | 53.2 ± 12.9 |
| Males, *n* (%) | 39 (78%) |
| BMI, mean ± SD, kg/m$^2$ | 27.7 ± 5.7 |
| Heart rate, mean ± SD, beats/min | 82.6 ± 28 |
| Systolic (mmHg) | 124 ± 16 |
| Diastolic (mmHg) | 75 ± 13 |
| Diabetes, *n* (%) | 5 (10%) |
| Left bundle-branch block, *n* (%) | 9 (18%) |
| Atrial fibrillation, *n* (%) | 9 (18%) |
| 6MWT, mean ± SD, months | 498 ± 128 |
| NYHA I | 9 (18%) |
| NYHA II | 21 (42%) |
| NYHA III | 19 (38%) |
| NYHA IV | 1 (2%) |
| Family history of SCD or DCM, *n* (%) | 9 (18%) |

myocardial biopsy (Meder *et al*, 2017). Exclusion criteria were all secondary causes of DCM. A total of *n* = 50 consecutive patients fulfilling the requirements were included, and detailed clinical baseline characteristics are summarized in Table 1.

NGS was performed on an Illumina platform, and libraries were generated from biopsy and blood using the TruSeq technology. For WGS, the library inserts are small spread, averaged on 296 bp (224–327 bp), which is necessary to fulfill technical requirements to reliably call large deletions, duplications, or inversions with high confidence (Fig 1B), especially in genomic regions of low sequence complexity and regions with difficulties for sequencing and mapping. Deep sequencing gained a coverage ranging from 40× to 66× with an average of 58× ± 6.5. Figure 2A shows the uniformity of the generated sequencing data over all chromosomes (blue line).

**High-resolution map of genomic structural variants in DCM**

The high-coverage WGS of DCM patients was subsequently subjected to paired-end mapping and split-read detection algorithms for the identification of SVs. The algorithms are implemented in Delly SV calling, an approach also employed in the latest release of the 1000 Genomes Project (Rausch *et al*, 2012; Sudmant *et al*,

2015). Inclusion of deletions and duplications with respective read-depth and inversions with support for both breakpoint and complex events resulted in 2,955 high confidence deletions, 797 duplications, and 146 inversions and complex structural variants in the 50 patients. SVs from small to large events in a range of 212 bp to 12.2 Mb (Fig 2B) were detected (deletions: 249 bp to 12.2 Mb, duplications: 212 bp to 1.7 Mb, and inversions and complex structural variants: 216 bp to 44.9 kb). Deletions and duplications were relatively uniformly distributed over all chromosomes and showed an expected modest accumulation in low complex regions, the telomeres, and centromeres (Fig 2A). To further increase confidence on the quality of the SV calls, we utilized raw intensity data from Illumina 450K methylation arrays measurements from the same samples (decreased signal intensities for deletions and increased intensities for duplications) (Feber *et al*, 2014; Meder *et al*, 2017). From a total of 633 SVs, in which sufficient methylation sites were available, we could independently validate the majority of deletions and duplications (*P*-value ≤ 0.05), showing the high quality of the variant calls (Fig EV1A), which was also underlined by PCR validation of randomly selected SV events (Fig EV1B).

To our knowledge, we present the most detailed map of genetic variation in a well-phenotyped cardiomyopathy cohort and find that the genome-wide SVs delete important loci directly related to gene expression, such as enhancers (*n* = 875), TFBSs (66,147), and lncRNAs (3,100) (Fig 2C). The fluctuating numbers of these affected regulatory loci in each patient are also visualized and can be mainly explained by the presence of some very large deletion events in some individuals. Most of the SVs reside within non-coding, intergenic regions. However, we find in total 4,305 exons (1,367 genes) to be entirely deleted by a hetero- or homozygous deletion, together with 630 deletions to be located inside an intron (Fig 2D). Duplications in DCM affect 132 enhancers, 9,674 TFBSs, 658 lncRNAs, and 389 protein-coding genes with 319 exons and 199 duplicated introns (Fig 2C and D).

Deletions of protein-coding exons result in average in a significant downregulation of the transcript (median expression value 0.8 for SV carriers) (Fig 2D, violin plots). Duplications do not show significant changes over all events, most likely due to the incomplete event not carrying the complete promoter or enhancer structure of the host gene. The effect of different SV-allele counts of a deletion on RNA-seq profiles is exemplarily shown in Fig 2E. In this case, the deletion leads to a reduction in exon expression of heterozygous SV carriers (orange line) and a complete absence of RNA expression in the deleted region of homozygous patients (red line, Fig 2E). 23.7% of the protein-coding genes hit by a SV were entirely covered by the

**Figure 2.  Structural variants (SVs) in DCM cohort.**

A  Genomic location of SVs are plotted in the Circos (Krzywinski *et al*, 2009) plot in the outer panel. Deletions face inwards and duplications outwards from the black circle as red lines. The inner panel displays the WGS coverage of each of the 50 patients (blue).

B  The size distribution for deletions (blue) and duplications (yellow) is plotted.

C  Genome-wide analysis of SVs shows the number of regulatory genomic elements to coincide with SVs. The number of regulatory genomic elements affected by SVs per patient is shown in the spider plots below.

D  Genome-wide analysis of SVs shows the number of protein-coding genes, exons, and introns to be affected by SVs. Expression ratios of deleted and duplicated protein-coding exons between SV carrier and wild type are shown in the violin plot. White circles represent median expression ratio, the box limits represent the 1$^{st}$ and 3$^{rd}$ quartile and the error bars represent the 1.5× interquartile range.

E  Example of the effect of a deletion on the expression of the associated transcript. The genotypes are coded by the colored lines (green: wild type, orange: heterozygous carrier, red: homozygous carrier). Deletions are depicted in blue and duplications in orange.

# Structural Variants in DCM cohort

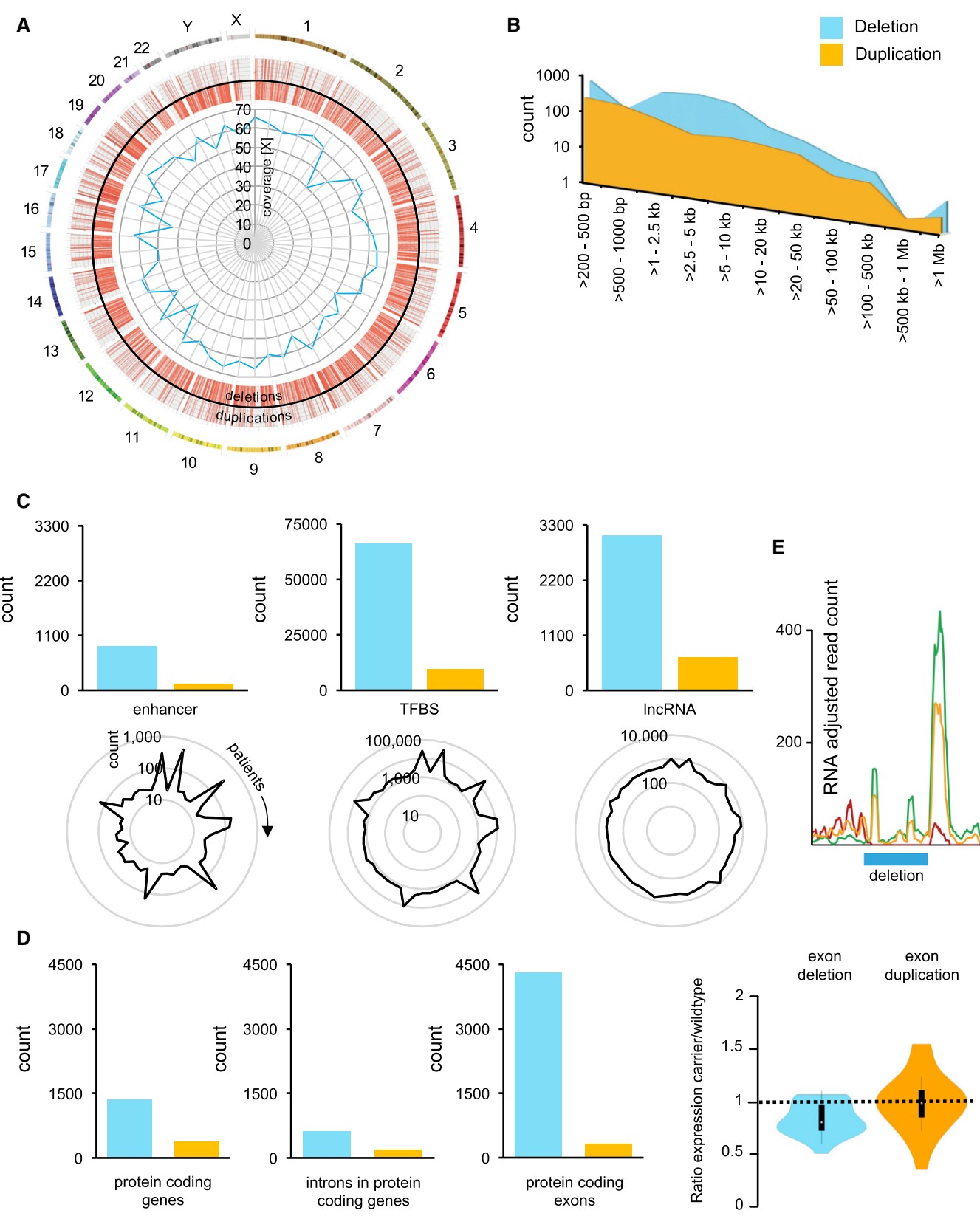

**Figure 2.**

**A**

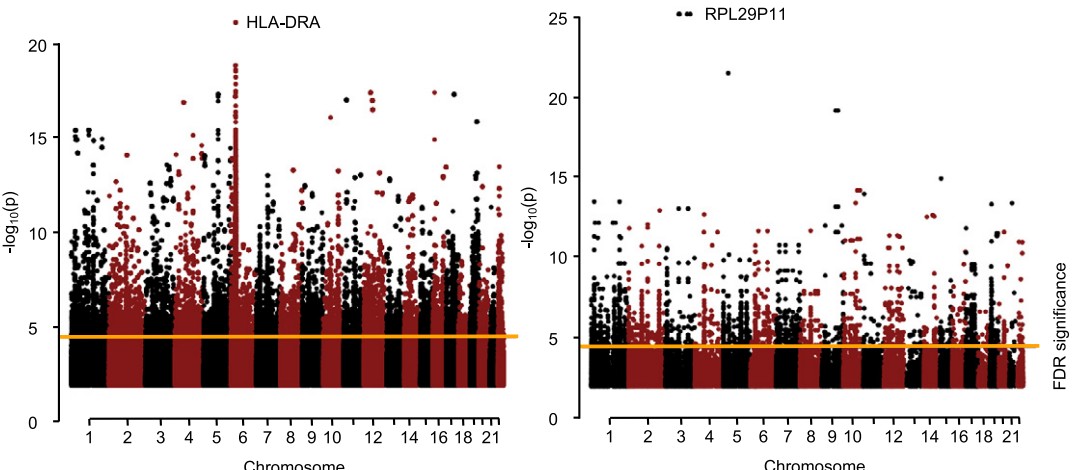

SNV-eQTL in LV-biopsy of DCM

SV-eQTL in LV-biopsy of DCM

**B**

Directional effect sizes of SV-eQTL and SNV-eQTL

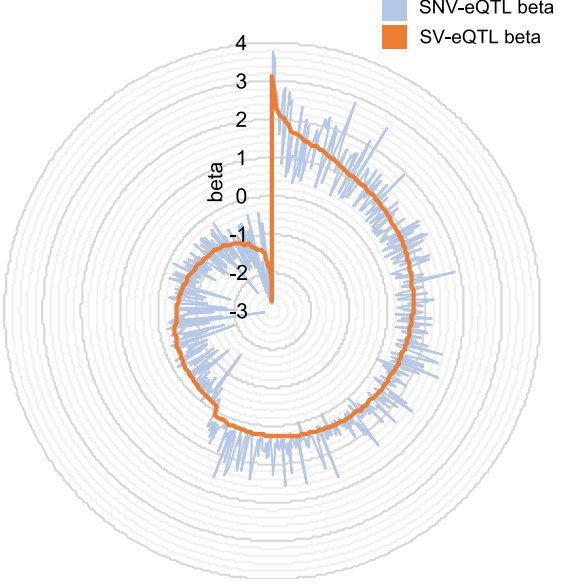

■ SNV-eQTL beta
■ SV-eQTL beta

**Figure 3. Genomic variant—eQTL.**

A   Manhattan plots showing the negative log$_{10}$ *P*-values from SNVs-eQTL or SVs-eQTL analysis. Orange lines indicate the significance threshold after correction for multiple testing (FDR ≤ 0.05).

B   The effect sizes (beta) of QTLs identified in a 1-Mb range from the SV-eQTL and SNV-eQTL analysis were aggregated, and the median for each gene is plotted.

**Single nucleotide and structural variants are linked to myocardial transcript dysregulation in DCM**

With the available high-quality mRNA expression data (Fig EV2), we performed SNV-eQTL analyses and found altogether 3,917 transcripts associated with a SNV (FDR significance level ≤ 0.05) (Fig 3A). The five most significant variant-expression associations are linking rs3129888, rs2239802, rs3135390, rs7196, rs2395182 with *Histocompatibility Antigen HLA-DR Alpha* (*HLA-DRA*). Other highly significant SNVs link to, for example, the *Endoplasmic Reticulum Aminopeptidase 2* (*ERAP2*) and *Transmembrane Protein 117* (*TMEM117*) (Appendix Table S1).

To test whether detected SVs also have a functional impact on the transcriptome as an intermediate cardiac phenotype, we next conducted a SV-eQTL analysis by performing correlation tests on the occurrence of all SV genotypes against all expressed transcript levels. Figure 3A is showing a genome-wide manhattan plot of all SV-eQTLs, with 2,652 reaching significance after correction for multiple testing (FDR *P* ≤ 0.05). When focusing on *cis*-regulation within 1 Mb distance (Sudmant *et al*, 2015), 75 genome-wide significant SV-eQTL links could be established (Appendix Table S2, Fig EV3) for 71 genes. Of those SV-eQTLs, ten expressed genes are altered due to the direct overlap with the SV, seven when considering linkage disequilibrium extension and another 54 in 1 Mb distance.

Interestingly, most of the eQTLs are exclusively expressed in heart tissue (left panel) and not in blood (right panel), suggesting a predominant cardiac effect. In 49 out of the 75 SV-eQTLs (65.3 %), a SNV-mediated effect is unlikely since no significant SNV-eQTLs are found in this region. Of the overlapping 760 eQTLs, the directional effect sizes of an additive linear model were comparable (Fig 3B). Also of note, we find 37 of the SV-eQTLs to affect non-coding RNAs, emphasizing the importance of the presumable regulative elements of untranslated regions in the human myocardial genome in DCM, which might affect other portions of the transcriptome. Such a complex, interacting system is supported by the differential effects

genetic variant. In detail, 377 genes were completely deleted at least on one of two chromosomes (126 genes are on the X-chromosome affected by a large deletion that is known from population studies), and 40 genes were completely duplicated. Overall, our results show that each individual carries more than 720 SVs on average.

## A    Gene regulation depending on elements hit by cis-SV-eQTLs

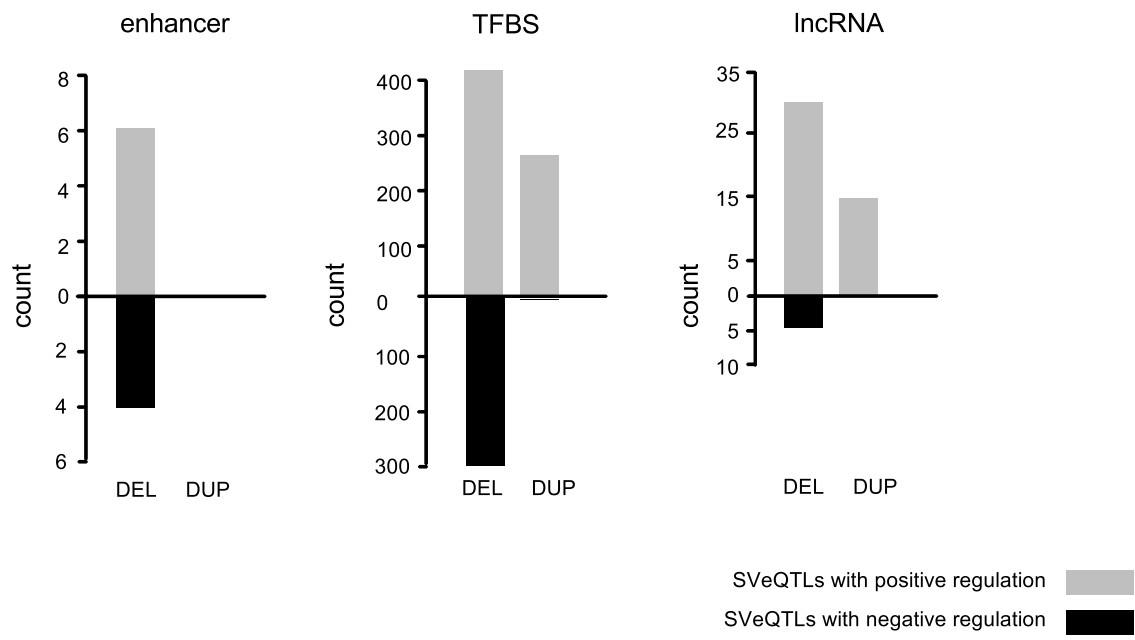

SVeQTLs with positive regulation
SVeQTLs with negative regulation

## B    Effect of common SVs on cardiac expression in DCM

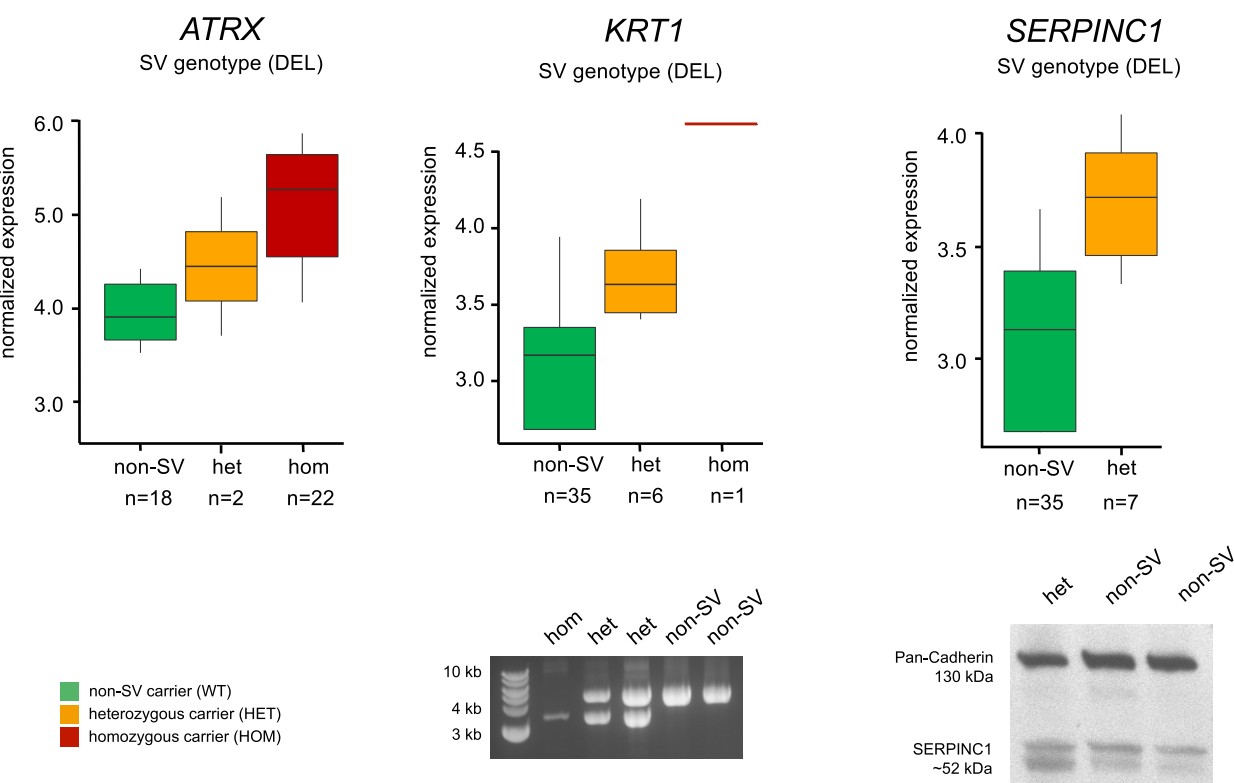

non-SV carrier (WT)
heterozygous carrier (HET)
homozygous carrier (HOM)

**Figure 4.**

observed for SV-eQTLs lying in enhancers, transcription factor binding sites, or non-coding RNAs (Fig 4A).

The allele frequencies of detected SVs and SV-eQTLs ranged from private (only detected in one individual) to common (frequency ≥ 5% of the cohort). Figure 4B exemplarily shows the effect of common SVs on the cardiac transcription; for example, an upregulation of the gene expression was found in the chromatin remodeler gene *Alpha Thalassemia/Mental Retardation Syndrome X-Linked* (*ATRX*), *Keratin 1* (*KRT1*), involved in activation of the immune system and beta catenin signaling, and *Serpin family C member 1* (*SERPINC1*), involved in blood coagulation. The increased amount of mRNA transcript for *SERPINC1* detected in SV carriers resulted in higher protein expression when compared to non-SV carrier (Fig 4B), indicating the profound effect to be expected by this class of variation.

Other SVs directly affected the coding transcript either by falling in the exon or introns. Figure EV4 shows altered expression of *Amidohydrolase Domain-Containing Protein 1 (AMDHD1)* gene that is implicated in amino acid synthesis. As shown, the SV results in an intronic deletion, which directly affects the expression of the mutant allele. To investigate the quantitative effect of the linked SV on the transcription in the corresponding 1-Mb interval, we calculated the determination of coefficient $R^2$ per SV event, showing the proportion of differential expression that can be explained by the corresponding SV. For the SV-eQTL *AMDHD1*, 18% of the total expression variation in the genomic interval can be explained by this approach. Another locus significantly affected by an SV in multiple patients (56%) is harboring five SV-eQTLs (Fig 5A). The 37-kb large deletion, which is spanning the entire *Glutathione S-Transferase Theta 2B* (*GSTT2B*) gene and the first two exons of *D-Dopachrome Tautomerase-Like (DDTL)* gene, accounts for 13% of the total expression alterations in this locus (Fig 5B). The methylation intensities for CpG sites located in this deletion showed decreased signals for heterozygous and homozygous SV carriers, respectively (Fig 5C). Very interestingly, of the five significantly SV-linked genes, two (*GSTT1 & 2*) are dysregulated in a mouse model of transverse aortic constriction (TAC) induced heart failure (Fig 5D), at least indicating the involvement in adaptive or maladaptive cascades in the failing mammalian heart.

While common SVs may have a regulatory effect on the transcriptome but theoretically only smaller effects on detrimental phenotypes, rare genetic variations bear the potential to significantly impact on human disease. Using SV-eQTL analysis for rare SVs involving gene regulatory regions, we found regulation of *PRELID1P4*, where a deletion resulted in an upregulation in two SV carriers (Fig 6A). For *ZNF35* a complex event, a combination of an inversion and a duplication in two different DCM patients led to

decreased transcript levels. Another complex SV (inversion and deletion) upregulated the expression of *CAMK2A* (Fig 6B), the gene for which the first knockout mouse was made (Silva *et al*, 1992). For validation, the genomic region of the complex SV (Fig 6D) was amplified and both single alleles (SV and non-SV) of a SV carrier were subjected to long-read nanopore sequencing, which confirmed the complex SV (Fig 6D). A more detailed analysis of the genetic context revealed that the *CAMK2A* upregulation on transcription level is exclusively driven by the expression of six C-terminal exons of the 19 exon spanning full-length *CAMK2A* (Fig 6B). Here, the SV explains 6% of the transcriptional variation of *CAMK2A* (Fig 6C) and 14% of the depicted genomic region.

## Transcriptome-wide effects of SVs in the heart

To examine the global impact of structural variation on myocardial gene expression in humans, we next covered both *cis* and *trans* regulatory mechanisms. Of the total of 20,712 genes found to be expressed in the myocardium, 16,449 genes (79%) could be directly or indirectly linked to a SV event at least in one proband (nominal $P \leq 0.01$). To determine to what extent those SV-eQTL genes contribute to the overall transcriptional variation found in the 20,712 genes, we applied linear modeling of attributed expression variation. By this approach, we inferred the degree of global SV-associated expressional variation to be 7.5% (Appendix Table S3). A similar picture is seen for the DCM-related genes, where SV-eQTLs were found to explain a fraction of 10.1% of the transcriptional variation.

# Discussion

The homeostasis of the human myocardium is tightly regulated involving several cellular mechanisms. Once disturbed by hemodynamic, metabolic, or inflammatory stimuli, cardiac remodeling is triggered to compensate for a molecular misbalance. If such adaptive processes fail, for example, due to pathogenic gene mutations, heart failure is the ultimate endpoint in a self-perpetuating, pathogenic vicious circle (Heusch *et al*, 2014). This might also be relevant for heart failure research, for example, in model systems as shown by a recent example of *Nnt* gene variants in bl/6 mice strains that completely modify the occurrence of heart failure due to stressors (Nickel *et al*, 2015).

While the role of SNPs is quite well established in heart failure of different causes, virtually nothing is known about the contribution of structural genomic variations on myocardial gene expression in human disease states. To our knowledge, we present the first study on genome-wide structural variations (SVs) in human heart disease

---

◀ **Figure 4. Gene regulation depending on elements hit by *cis*-SV-eQTLs.**

A  Shown is the number of regulatory genomic regions hit by SV-eQTLs leading to a positive (gray) or negative (black) gene regulation. Annotation is based the most abundant transcript per gene based on the RNA-seq data.

B  Effect of common SVs on cardiac transcription in DCM: Shown are the gene expression levels in patients carrying a linked heterozygous SV (orange) or homozygous SV (red) in comparison with the patients not carrying the corresponding SV (green). Structural variants linked via eQTL to the cardiac gene expression of *ATRX*, *KRT1* and *SERPINC1* are frequently detected in DCM patients (9–58%). Horizontal lines represent the median normalized expression, the box limits represent the 1st and 3rd quartile and the error bars represent the 1.5× interquartile range. PCR-based analysis confirms the structural variant linked to *KRT1* expression (see also Fig EV1B for additionally validated SVs). Immunoblot analysis of Serpinc1 confirms increased protein levels in myocardium of a SV carrier, as predicted by the SV-eQTL.

Source data are available online for this figure.

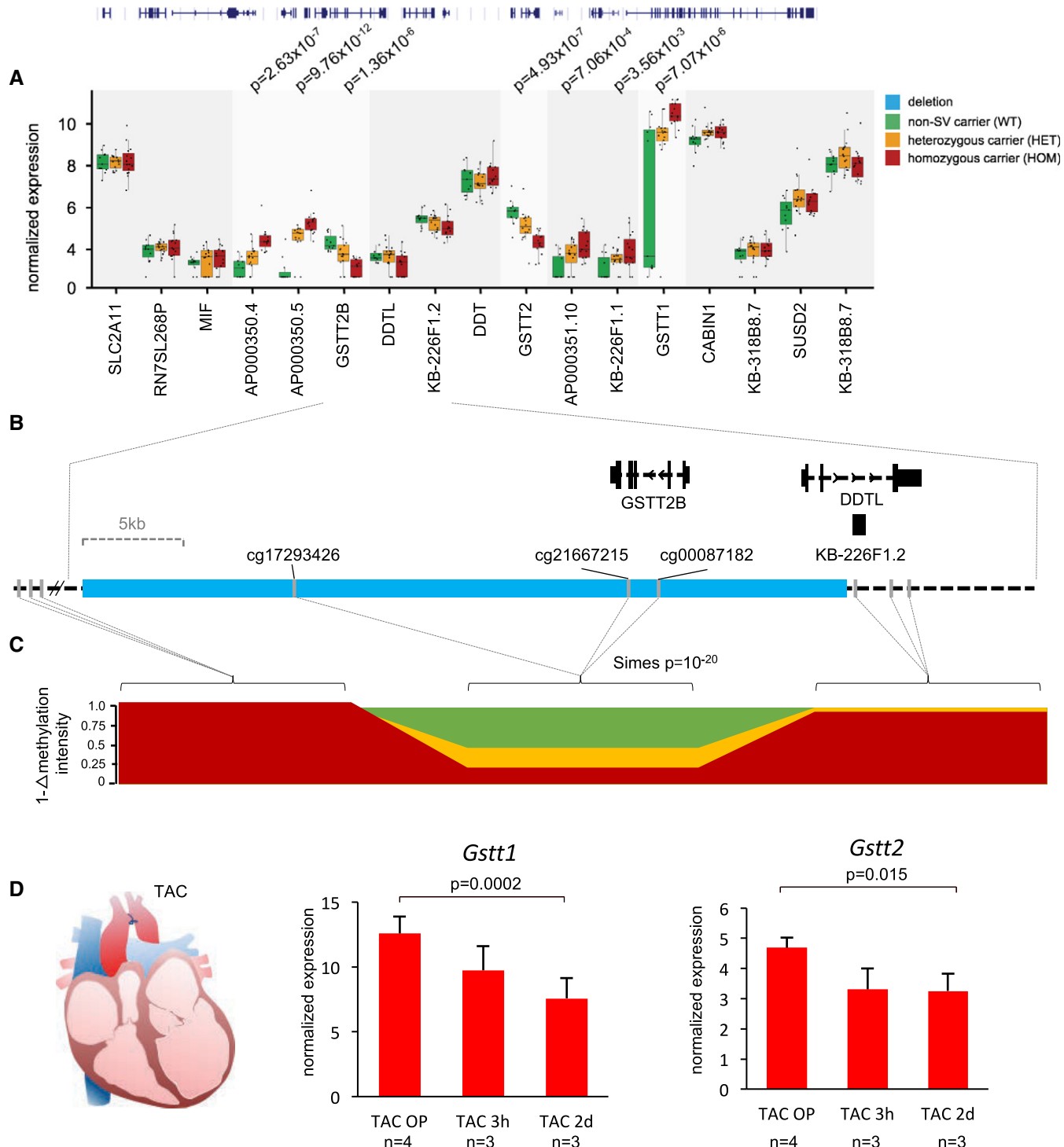

**Figure 5.  Potential relevance of SV-eQTLs in adaptive or maladaptive pathways in heart failure.**

A   A 37-kb deletion in the glutathione S-transferase theta (GSTT) locus affects multiple genes in linkage disequilibrium. Horizontal lines represent the median normalized expression, the box limits represent the 1st and 3rd quartile and the error bars represent the 1.5× interquartile range.

B   Genes are directly deleted by the SV, as is for *GSTT2B*, or partly deleted, as is for *DDTL*.

C   The validation of the linked deletion by methylation profiling reveals significantly reduced probe intensities for homozygous and heterozygous SV carrier compared to non-carriers.

D   For *GSTT1* and *GSTT2*, a significant dysregulation can be detected in a heart failure (TAC) mouse model at different time points post-TAC surgery, indicating that the genes may be functionally relevant in the setting of heart failure. Error bars represent ±SD; at baseline (TAC OP) values represent the mean of four biological replicates and three hours after TAC (TAC 3h) and two days after TAC (TAC 2d) values represent the mean of three biological replicates. *P*-values represent significance levels from Student's *t*-test.

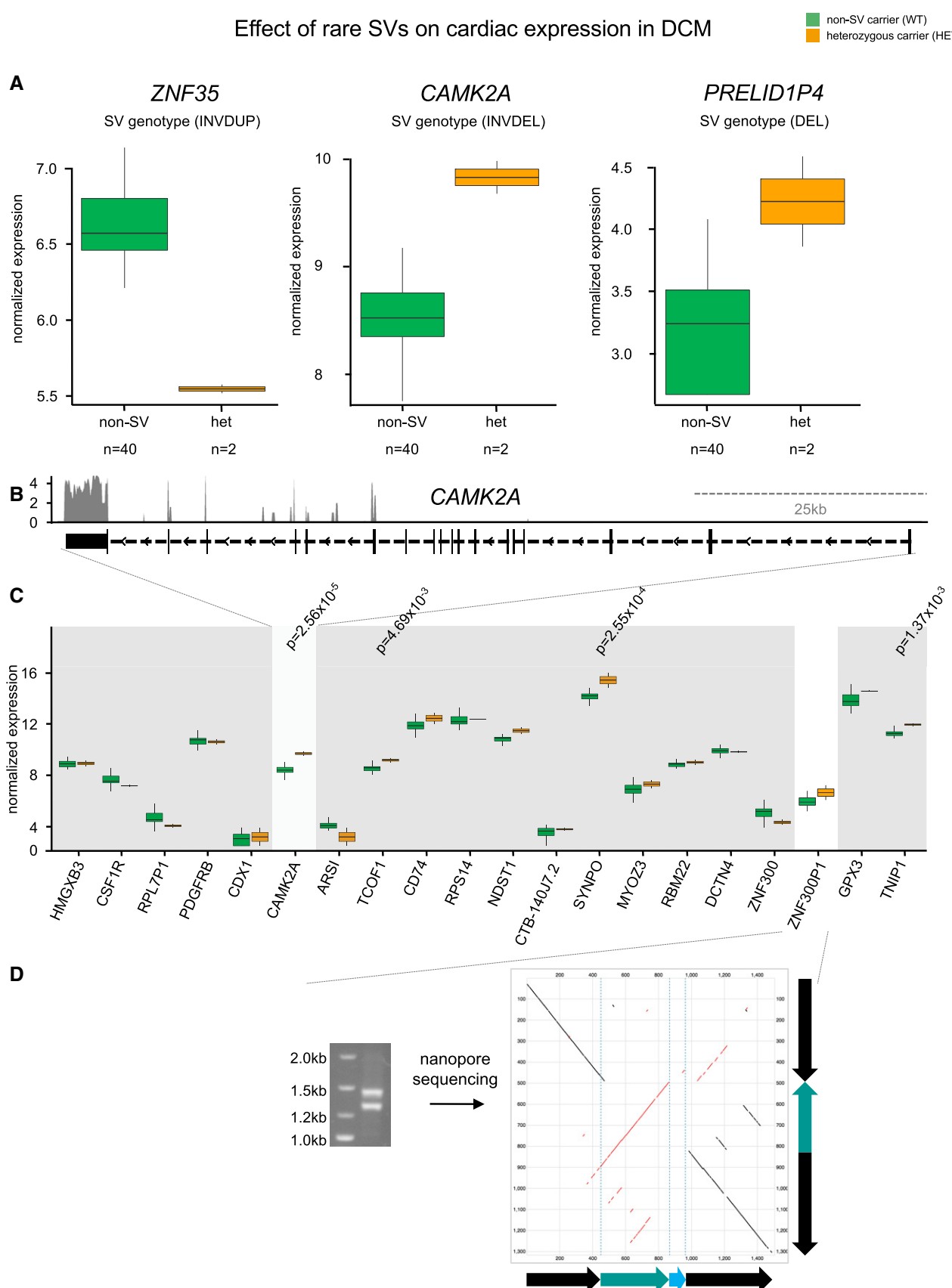

Effect of rare SVs on cardiac expression in DCM

Figure 6.

**Figure 6.  Effect of rare SVs on cardiac transcription in DCM.**

A   Shown are the gene expression levels in patients carrying a linked heterozygous SV (orange) in comparison with the patients not carrying the corresponding SV (green). Rare SVs are detected in only < 5% of DCM patients and alter myocardial *PRELID1P4*, *ZNF35* and *CAMK2A*, expression. Horizontal lines represent the median normalized expression, the box limits represent the 1st and 3rd quartile and the error bars represent the 1.5× interquartile range.

B   A short *CAMK2A* transcript is expressed in human myocardium.

C   The complex SV linked to CAMK2A expression is located in the intron of a *cis*-gene. Horizontal lines represent the median normalized expression, the box limits represent the 1st and 3rd quartile and the error bars represent the 1.5× interquartile range. *P*-values are obtained from linear regression model.

D   The genomic context of a heterozygous SV carrier was amplified and analyzed using single molecule nanopore sequencing. Shown is the alignment against the reference sequence, detailing the inversion-deletion event in single DNA molecules.

and show the high abundance and general effects of SVs in DCM using a multi-omics dataset. To precisely dissect the role of these poorly investigated mechanisms, we combined state-of-the-art whole-genome sequencing (WGS) and genome-wide transcription profiling (RNA-seq) from blood and interventionally obtained myocardial biopsies. From this integrated dataset, we were able to show a highly relevant effect of SVs on myocardial gene expression.

Since the first draft of the Human Genome project was published (Lander *et al*, 2001), the awareness of the complex and well-regulated nature of the genome has become apparent. Identification of regulatory elements, such as lncRNAs acting on the expression of genes, and the growing perception that most untranslated, formerly called "junk DNA", can be functional and is biochemically active (ENCODE Project Consortium, 2012) has shifted the focus from coding point mutations to structural aberrations in the entire coding and non-coding regions of the genome. Structural variants, including deletions, duplications, and complex events, contribute enormously to the phenotypic complexity of mammalian organisms, covering more varying nucleotides between individuals than SNVs (Sudmant *et al*, 2015).

The lack of evidence for the contribution of SVs to heart disease is not only conceptional, but also has technical reasons. In the past, fluorescence *in situ* hybridization or array-based techniques were used to infer copy number variants as one class of SV. However, when comparing these techniques to recent studies using WGS, the low resolution and high imprecision of those methods became apparent (Yoon *et al*, 2009). In the current investigation, we relied on deep coverage WGS that was validated by array-based techniques and in selected cases PCR-based and long-read nanopore sequencing. This resulted in a precise mapping of genomic alterations and even complex SV that stem from combined inversion–deletion–duplication events. By using mRNA expression analysis, we could link the SVs to a functional effect that is for many SVs restricted to the heart.

To systematically identify SV-associated aberrant expression of cardiac genes, we performed both, SNV and SV-eQTL analyses. In total, 75 SV-eQTLs were identified to be significantly associated with cardiac gene expression changes, from which at least 65.3% are not explained by SNVs. Of these SV-eQTLs, several genes are involved in key pathways of cardiomyopathies or are associated with DCM. For instance, the expression of the *chloride Intracellular channel 2 protein* (*CLIC2*) has been previously found to be downregulated in DCM (Molina-Navarro *et al*, 2013), but we now link this transcript to larger deletions that are present in approximately one-third of our DCM patients. CLIC2 is a negative regulator of the ryanodine receptor channel RYR2 and thereby not only acts on chloride homeostasis, but also $Ca^{2+}$ release (Dulhunty *et al*, 2005). Point mutations

in *CLIC2* were furthermore described as rare X-linked channelopathy leading to DCM (Takano *et al*, 2012). Our findings not only suppose that CLIC2 is a modifier for DCM, and it also highlights the genomic diversity in important ion channels that are intentionally or as side effects targeted by anti-arrhythmic drugs. *CAMK2A*, as an example of a rare SV-eQTL, is a serine/threonine kinase belonging to the calcium/calmodulin-dependent protein kinase superfamily and has in numerous studies been shown to modulate calcium handling and signaling, for example, Wnt signaling, and is associated with different forms of cardiomyopathies (Little *et al*, 2009; Toko *et al*, 2010; Zhang *et al*, 2015). In general, CAMK2 components are able to phosphorylate the full-length cardiac titin and hence modulate the sarcomeric stiffness (Hidalgo *et al*, 2013). In full-length transcripts, which are mainly present in neuronal tissue, the C-terminal domain is thought to additionally contain a localization peptide directing the functional enzyme to its site of action, the nuclear membrane, or the sarcoplasmic reticulum. The here detected shorter transcript encodes for the self-association domain responsible for assembling CAMK2 subunits to a fully functional multimer (Bayer *et al*, 1996). Aberrant expression of this transcript might disturb this molecular mechanism.

Whereas most of the discussed examples show an impact of SVs on protein-coding genes, the WGS dataset also revealed a large portion of the SVs to be located in the non-protein-coding region of the genome. Here, lncRNAs were the second most frequently affected class of regulatory elements in the significant SV-eQTLs. Although research on lncRNAs is still in its beginnings, convincing evidence for their contribution on the transcriptome homeostasis in the cardiovascular system in health and disease has been shown (Uchida & Dimmeler, 2015; Wang *et al*, 2016). Similar to miRNAs, lncRNAs bear the potential to treat muscle disease by re-establishing gene regulatory networks (Matsui & Corey, 2017).

The described findings underline the high potential of SVs to act on cardiac gene expression in a multifaceted manner (Zhang & Lupski, 2015). Chiang *et al* (2017) recently estimated that 3.5–6.8% of all *cis*-eQTLs are driven by a SV. This immediately raises the question about their pathophysiological role in DCM. We performed stringent statistics and focused on *cis* regulatory effects within linkage disequilibrium to provide robust estimates on the relevance of SVs in this context. When including *trans* regulation, as much as 7.5% of the whole myocardial expression variation could be explained by SVs and 10.1% of the variation of DCM-related genes (KEGG). Besides this statistical evidence of association, we, however, cannot proof a causal relationship with our study design. Subsequent investigations on the functional role of the identified targets need to be performed in line with large-scale studies including excellently phenotyped control cohorts. With the knowledge

about the impact of SVs on myocardial gene patterns, it seems reasonable to genetically characterize patients selected for innovative targeted therapies that rely on modification of the cardiac transcriptome by either miRNA, lncRNAs, or gene repair.

# Materials and Methods

## Patients and study design

The characterization of samples and patient data has been approved by the ethics committee, medical faculty of Heidelberg, participants have given written informed consent, and Care4DCM project was conducted (Meder *et al*, 2017). Symptomatic DCM patients were consecutively, prospectively enrolled. A prerequisite for enrollment was leftover myocardial tissue from the routine diagnostic workup. For the exclusion of secondary causes of DCM, all patients underwent diagnostic coronary angiography, histopathology of myocardial biopsies, echocardiography, cMRI, comprehensive clinical phenotyping, and biomarker measurements. Patients with valvular or hypertensive heart disease, history of myocarditis, regular alcohol consumption, or cardio-toxic chemotherapy were also excluded.

Biopsy specimens were obtained from the apical part of the free left ventricular wall (LV) from DCM patients undergoing cardiac catheterization using a standardized protocol. Biopsies were rinsed with NaCl (0.9%) and immediately transferred and stored in liquid nitrogen until DNA or RNA was extracted. Total RNA was extracted from biopsies using the RNeasy kit according to the manufacturer's protocol (Qiagen, Germany). RNA purity and concentration were determined using the Bioanalyzer 2100 (Agilent Technologies, Berkshire, UK) with a Eukaryote Total RNA Pico assay chip.

## Whole-genome sequencing

1 µg of total gDNA was sheared using the Covaris™ S220 system, applying two treatments of 60 s each (peak power = 140; duty factor = 10) with 200 cycles/burst. 500 ng of sheared gDNA was taken, and whole-genome libraries were prepared using TruSeq DNA sample preparation kit according to manufacturer's protocols (Illumina, San Diego, US). Sequencing was performed on an IlluminaHiSeq 2000, using TruSeq SBS Kit v3 and reading two times 100 bp for paired-end sequencing, on four lanes of a sequencing flowcell.

Demultiplexing of the raw sequencing reads and generation of the fastq files was done using CASAVA v.1.82. The raw reads were then mapped to the human reference genome (GRCh37/hg19) with the burrows-wheeler alignment tool (BWA v.0.7.5a) (Li & Durbin, 2009), and duplicate reads were marked (Picard-tools 1.56) (http://picard.sourceforge.net/). Next, we used the Genome-Analysis-Toolkit according to the recommended protocols for variant recalibration (v. 2.8-1-g932cd3a) and variant calling (v.3.3-0-g37228af) as described in the respective best-practices guidelines (https://www.broadinstitute.org/gatk/guide/best-practices) (DePristo *et al*, 2011). Structural variants were detected using paired-end and split-read algorithms. For this, aligned reads were processed with Delly (v0.6.3) (Rausch *et al*, 2012), excluding centromeric and telomeric regions of hg19 and employing an insert size cutoff of median + 9 times median absolute deviation for deletions. SVs were included in the analysis, if the median normalized read-depth ratio of heterozygous SV carriers compared to SV non-carriers was < 0.8 or > 1.3 for deletions or duplications, respectively. Only those inversions were kept in the analysis that had inversion-supporting paired ends for both breakpoints. Complex structural variants were also discovered by Delly applying two complementary paired-end clusters and a concurrent read-depth decrease or increase for an inversion and deletion (INVDEL) or inversions and duplication (INVDUP), respectively. Obtained SVs were re-genotyped in the l WGS data of the 1000 Genome Project the phase 3 (http://www.1000genomes.org/) to obtain their allele frequencies.

## Expression analysis

From the 50 samples in the cohort, high-quality RNA-seq libraries from left ventricular RNA could be generated for 42 patients using TrueSeq RNA Sample Prep Kit (Illumina). Sequencing was performed 2 × 75 bp on a HiSeq2000 (Illumina) sequencer. Samples were sequenced to a median paired-end read count of 31.5 million (range: 3.1–99.9). Unstranded paired-end raw read files were mapped with STAR v2.4.1c (Dobin & Gingeras, 2015) using GRCh37/hg19 and the Gencode 19 gene model (http://www.gencodegenes.org/). Only uniquely mapped reads were counted into genes using subread's featureCounts program (Liao *et al*, 2014) (subread version 1.4.6.p1), and mapping percentages were median 87.8 (range: 23.0–90.9). Prior to statistical analyses, genes with very low expression levels (average reads ≤ 1, detected reads in less than 50% of the samples) were removed resulting in 20,712 significantly expressed genes. Count data were normalized by rlog normalization (Love *et al*, 2014), which is an improved method of the variance stabilization transformation (Anders & Huber, 2010) as recommended for eQTL by the original MatrixEQTL publication (Shabalin, 2012).

## SV-eQTL

An eQTL analysis between SVs and gene expressions is performed on the 42 patients with high-quality transcriptome data from biopsy samples. MatrixEQTL (Shabalin, 2012) is used to correlate the 3,897 SV events and the expression profiles of 20,712 genes. To account for LD effect, SV spans are first extended using a twofold extension method as such. The SV is extended to the range from the furthest upstream base pair with LD $R^2 > 0.5$ to the furthest downstream base pair with $R^2 > 0.5$ to the SV. Then, the range is further extended to the next immediate recombination sites. LD and recombination site data for GRCh36 were first obtained from Hapmap.org and lifted over to GRCh37 using UCSC liftover tool.

For performing an eQTL, it is important to define an interval where a possible link is calculated. To estimate a genuine window between gene and extended SV, we chose to follow methods used in the latest release (phase3) of the 1000 Genomes croject, where an eQTL was considered within a 1 Mb range (Sudmant *et al*, 2015). Due to the genomic complexity of the human leukocyte antigen locus (HLA), located within the 6p21.3 region on the short arm of

human chromosome 6, we excluded these loci from the SV-eQTL analysis.

## Coefficient of determination ($R^2$) analysis

From each SV-eQTL with *P*-value of 1% or less, a residual variance was computed. For a set of SV-linked genes, the residual variance of each gene was summed to generate a residual variance of the set of genes. Together with the total variance of the set, the coefficient of determination of the set of genes was then calculated accordingly, that is, $1 - \sum$ (residual variances)/(total variance). The value represents then the proportion of total variance that the model explains. The coefficients of determination are meant to be descriptive, and hence, no associated statistical significances are calculated.

## SNV-eQTL

An eQTL analysis between SNVs and gene expressions was performed on the 42 patients with high-quality transcriptome data. MatrixEQTL (Shabalin, 2012) is used to correlate the 14,720,818 SNV events and the autosomal expression profiles of 20,172 genes that are within 1 Mb base pairs of each other. To account for LD effect, SNV spans are first extended using a twofold extension method as such. The SNV is extended to the range from the furthest upstream base pair with LD $R^2 > 0.5$ to the furthest downstream base pair with $R^2 > 0.5$ to the SNV. Then, the range is further extended to the next immediate recombination sites. LD and recombination site data for GRCh36 were first obtained from Hapmap.org and lifted over to GRCh37 using UCSC liftover tool.

## Verification of structural variants using high-density DNA methylation arrays

It has been shown that Illumina 450k methylation assay can be used to profile copy number alterations since the overall signal intensity of the methylated and unmethylated probes reflects the DNA amount and thereby copy number (Feber *et al*, 2014; Meder *et al*, 2017). To verify structural variants using MatrixEQTL, we correlated 633 SV events that covered at least one methylation locus measured on the Illumina 450k chip and the overall signal intensity for the respective methylation loci in blood and cardiac tissue using tissue as covariate. In case of duplications, we tested for increased signal and in case of deletions for reduced intensity in the presence of events. For each SV event, an aggregate significance level was obtained using the *simes* procedure (Rødland, 2006).

## Polymerase chain reaction for SV validation

For technical validation of SVs, the PrimeSTAR GXL DNA Polymerase (TaKaRa Bio inc., Tokyo) was used, taking 10 ng of gDNA as template in 20 μl reaction volume using the following primers: XKR9 forward (5′-TTGTGTCCTAGACAGGCGAGTG-3′) and XKR9 reverse (5′-GCCAAATGAGGAGCTTGGCAAT-3′). TNKS2-AS1 forward (5′-TAGTACAGCTGCCCCTTGTGAC-3′) and TNKS2-AS1 reverse (5′-TGGCAGCCTGTTTAGATCCACT-3′). KBTBD11-OT1 forward (5′-ACAAGCGCTTTCAGGGGAAATG-3′) and KBTBD11-OT1 reverse (5′-TTTGGGTGAAGGCGTCTAACCA-3′). KRT1 forward (5′-GGGCGTGGATTC

**The paper explained**

**Problem**
The myocardium has to permanently adapt to changes in the hemodynamic demand, aging of the organism and multiple external stressors. Opposing to the dynamic nature of these mechanisms are static effects on the transcriptome originating from genetic variation. Here, we investigated the presence of complex structural genomic variations in patients with DCM and performed genetic association analysis using a multi-omics strategy.

**Results**
Most of the detected and validated SVs affect non-protein-coding regions of the genome, resulting in transcript mis-expression by *cis*-regulation. The effect sizes of SV-eQTLs are similar to those found for single nucleotide variants, and many are specific for heart tissue compared to peripheral blood. By this genome-wide strategy, we could identify several interesting candidate loci that are likely involved in myocardial (mal)adaptation.

**Impact**
The findings highlight the role of SVs in myocardial gene expression regulation and require genome sequencing for patient-specific approaches targeting the cardiac transcriptome.

TTGTTCACAG-3′) and KRT1 reverse (5′-GTCTAACTTGGGGGTACGTGCT-3′).

## Long-read nanopore sequencing

Genomic intervals were amplified using forward (5′-CCGTAAGTGCAATGCAATCCCT-3′) and reverse (5′-CTCCAGCAGGGTCTGAGGTTAC-3′) primer with Qiagen (Germany) Taq polymerase and separated in 1% agarose gel stained with Midori Green Advance (Nippon Genetics Europe, Germany) and extracted from the gel. 1.25 μg of the variant and wild-type fragment were taken for the library preparation according to the manufacturers protocol (GDE_1002_v1_revB_17Nov2015) with the SQK-MAP006 sequencing kit on a MinION Mk1. Obtained FAST5 sequences were processed with poRetools (Loman & Quinlan, 2014) and aligned with bwa-mem (0.7.10-r789).

## Immunoblot analysis

Protein expression analysis of was performed on left ventricular material of independent samples, which were homogenized with ceramic beads (1.4 and 2.8 mm) in 400 μl lysis buffer (50 mM Tris–HCl, 120 mM NaCl, 5 mM EDTA, 0.1% NP-40, 1 mM DTT, 1 mM sodium metavanadate, 1 mM sodium fluoride, 0.2 mM PMSF, protease inhibitor (cOmplete tablet Roche cat# 04693116001) and phosphatase inhibitor (PhosSTOP Roche cat# 04906837001). Equal amounts of protein as tested by the Pan Cadherin control (abcam ab16505; dilution 1:1,000) were diluted in 4× Laemmli sample buffer (Bio-Rad cat# 161-0747) and separated on a 4–20% gradient gel (Bio-Rad, Germany, cat.# 4561094). Primary antibody for SERPINC1 (ThermoFisher Scientific, cat# PA5-13674; dilution 1:750) and secondary HRP-linked anti-rabbit IgG antibody (Cell Signaling Technology, cat#7074; dilution 1:2,500) were used. Pierce™ ECL Western Blotting Substrate (cat# 32209) was used for the detection of HRP.

## Transverse aortic constriction

Institutionally available TAC data were used to investigate potential dysregulation of SV-eQTL homologous transcripts in mice with induced heart failure. TAC was performed as previously described (Volkers *et al*, 2013). Briefly, in 8-week-old male mice, the aorta between the innominate and the left common carotid arteries was ligated with a 7-0 polypropylene suture and a 27-gauge needle, which was removed after ligation. Before extubation and closing the chest, the pneumothorax was reduced. A sham procedure in which the aorta was not bended was also performed.

## Statistical analyses and databases

Statistical analyses were carried out in R-3.2.2 (R Development Core Team, 2008). FDR correction of significance levels was performed using the Benjamini–Hochberg procedure (Benjamini & Hochberg, 1995). TFBSs employed in this study were annotated by http://genome.ucsc.edu/cgi-bin/hgTrackUi?db = hg19&g = wgEncodeRegTfbsClusteredV3. Enhancers used for annotation were obtained from FANTOM5 human permissive enhancers phase 1 and phase 2 (FANTOM Consortium and the RIKEN PMI and CLST (DGT) *et al*, 2014). All other genomic loci used for functional annotation were pulled from GENCODE (v19).

## Data availability

The data are freely accessible (accession number CMS-SV-17; https://ccb-web.cs.uni-saarland.de/cms).

**Expanded View** for this article is available online.

## Acknowledgements
This work was partially supported by grants from the German Ministry of Education and Research (BMBF: Project CaRNAtion), DZHK ("Deutsches Zentrum für Herz-Kreislauf-Forschung"—German Centre for Cardiovascular Research), the European Union (FP7 BestAgeing), and Siemens Healthcare GmbH (Siemens/University Heidelberg Joint Research Project: Care4DCM). We thank EMBL GeneCore and IT facilities at EMBL-EBI, EMBL-Heidelberg, and Sascha Meiers for technical support.

## Author contributions
BM, HAK, AEP, EW, MW designed the study; FS-H, EK, JOK, TW, DM, AC, MV, SB, DO, AA, DBH recruited patients and contributed data; JH, SM, KSF, RN, ER performed experiments; SM, JH, AEP, CD, AL, DP, TR, J-NB, DMB, AK analyzed data; BM, HAK, JH, SM wrote the manuscript.

## Conflict of interest
Andreas E. Posch, Carsten Dietrich and Maximilian Wuerstle are employees of Siemens Healthcare. Dietmar Pils was an employee of Siemens AG Österreich.

## For more information
Meder Lab: www.mederlab.com
Institute for Cardiomyopathies Heidelberg (ICH):
www.cardiomyopathie-heidelberg.de
Cardiac Multi-Omics Server: https://ccb-web.cs.uni-saarland.de/cms
Center for Cardiovascular Research: https://dzhk.de

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
