## [Review Process File · EMBO Molecular Medicine]

Genomic Structural Variations Lead to Dysregulation of Important Coding and Non-Coding RNA Species in Dilated Cardiomyopathy

Jan Haas, Stefan Mester, Alan Lai, Karen S Frese, Farbod Sedaghat-Hamedani, Elham Kayvanpour, Tobias Rausch, Rouven Nietsch, Jes-Niels Boeckel, Avisha Carstensen, Mirko Voelkers, Carsten Dietrich, Dietmar Pils, Ali Amr, Daniel B Holzer, Diana Martins Bordalo, Daniel Oehler, Tanja Weis, Derliz Mereles, Sebastian J Buss, Eva Riechert, Emil Wirsz, Maximilian Wuerstle, Jan O Korbel, Andreas Keller, Hugo A Katus, Andreas E Posch, Benjamin Meder

Corresponding author: Benjamin Meder, University of Heidelberg

Review timeline:

Submission date:	29 March 2017
Editorial Decision:	09 May 2017
Revision received:	11 August 2017
Editorial Decision:	26 September 2017
Revision received:	29 September 2017
Accepted:	05 October 2017

Transaction Report:

Editor: Roberto Buccione & Céline Carret

1st Editorial Decision

09 May 2017

Thank you for the submission of your manuscript to EMBO Molecular Medicine. We are sorry that it has taken so long to get back to you on your manuscript.

In this case we experienced unusual difficulties in securing three willing and appropriate reviewers. As a further delay cannot be justified I have decided to proceed based on the two available consistent evaluations.

In addition to the above, reviewer cross-commenting and additional extensive internal discussion were required to reach a final decision, further adding to the delay.

As you will see, both reviewers albeit with different degrees of enthusiasm, point to the same basic issues, i.e. that the study is essentially a descriptive one with inconclusive links between the SVs and DCM pathology. More in detail, (1) the comparison between DCM and control population (1000-genome project) is rather weak, and thus the potential causality of the SVs on DCM pathology cannot be reached, and (2) the causality with the eQTL analysis is also somewhat weak, especially for those SVs that do not impact coding parts of genes.

After further discussion and reviewer cross-commenting, it was agreed that to upgrade the

conclusiveness of the manuscript would require an amount of work that would extend well beyond a reasonable timeframe for a revision. On the other hand, it also clearly emerged that the manuscript does present useful general and novel informative value, susceptible of course to improvement based on their indications.

Overall, while it is clear that publication of the manuscript cannot be considered at this stage, we are willing to consider a substantially revised submission. Specifically, we would ask you to completely de-emphasise the part between DCM to general population comparison (unless you can fully address the very specific concerns raised on this aspect), and focus on the SV-eQTL analysis, to be further improved according to the reviewers' requests, including with additional experimental data where appropriate. The revised manuscript will undergo a second round of review.

Please note that it is EMBO Molecular Medicine policy to allow a single round of revision only and that, therefore, acceptance or rejection of the manuscript will depend on the completeness of your responses included in the next, final version of the manuscript.

As you know, EMBO Molecular Medicine has a "scooping protection" policy, whereby similar findings that are published by others during review or revision are not a criterion for rejection. However, I do ask you to get in touch with us after three months if you have not completed your revision, to update us on the status. Please also contact us as soon as possible if similar work is published elsewhere.

Please note that EMBO Molecular Medicine now requires a complete author checklist (<http://embomolmed.embopress.org/authorguide#editorial3>) to be submitted with all revised manuscripts. Provision of the author checklist is mandatory at revision stage; the checklist is designed to enhance and standardize reporting of key information in research papers and to support reanalysis and repetition of experiments by the community. The list covers key information for figure panels and captions and focuses on statistics, the reporting of reagents, animal models and human subject-derived data, as well as guidance to optimise data accessibility. The Author checklist will be published alongside the paper, in case of acceptance, within the transparent review process file.

Finally, we now mandate that all corresponding authors list an ORCID digital identifier. You may do so through our web platform upon submission and the procedure takes <90 seconds to complete. We also encourage co-authors to supply an ORCID identifier, which will be linked to their name for unambiguous name identification.

I look forward to seeing a revised form of your manuscript in due time.

***** Reviewer's comments *****

Referee #1 (Comments on Novelty/Model System):

This study provides the first identification of genetic structural variations (SVs) in patients with dilated cardiomyopathy (DCM). However, the data is descriptive and the authors fail to include a control group of patients without DCM in their analysis of SVs. I believe they also over-conclude what the pathological implications of SVs are in DCM patients, especially since they compare SVs between their DCM patients and previously published population SVs. These and other concerns are discussed in detail in the "Comments to Authors".

Referee #1 (Remarks):

This study analyzed what genome-wide structural genomic variants (SVs) occur in 50 patients with dilated cardiomyopathy (DCM). A total of 80,635 non-protein coding elements of the genome were found to be deleted or duplicated by an SV, including 1,756 protein-coding transcripts. This involved genes involved in chromatin remodeling, host defense, apoptotic processes, haemostasis, sarcoplasmic reticulum calcium release and calmodulin-dependent signaling. It is concluded that this is the first study to pinpoint the genetic variability due to SVs in DCM and demonstrate their specific molecular effect.

General Comments:

This is a detailed study that is the first study to demonstrate genomic variability due to SVs in DCM. It is competently performed study that provides information on a large number of SVs present in DCM patients. However, it is a descriptive study, and the authors over-conclude what the implications of these SVs have on DCM disease pathology. It is unfortunate that the authors did not include a control group of patients without DCM to determine which of the SCVs are specific for the patients with DCM. A comparison of population matched controls was performed by in silico re-genotyping of all identified SVs from the 1000 Genomes Project, but this is not the same as including their own control patients. This in silico analysis revealed 538 novel deletions and 194 novel duplications in human DCM patients.

Comparison of the SVs obtained in the DCM patients with the subjects of the 1000 Genomes Project identified a number of novel genes that are proposed to be involved in the pathogenesis of DCM, including ADCYAP1, ARHGAP1, DAXX, SERPIN1, EPX, CLIC2, and CAMK2A. While the authors discuss the potential link of these changes to DCM pathology, the actual link between alterations in these genes to DCM pathology is weak. It is also unfortunate that no data is provided as to whether the actual expression of the proteins associated with these genes is altered in DCM. This, of course, is complicated by the lack of a control subject group in this study.

It is not clear whether any of the proposed SVs seen in the genome result in adaptive or maladaptive changes in DCM. Therefore, the authors need to be cautious in linking these SVs to DCM pathology.

Specific Comments:

- 1) Abstract, last paragraph: The authors suggest that they pinpoint the genomic variability due to SVs and demonstrate their specific molecular effect. This is an overstatement and no specific molecular effects are demonstrated.
- 2) Introduction: para 1, line 3: The references cited do not discuss the cascade involving cardiac energy metabolism implied in the sentence.
- 3) Introduction, para. 2, line 1: The authors suggest that changes in gene expression are "adaptive". The possibility that they may be maladaptive is not discussed.

Referee #2 (Remarks):

This manuscript reports structural variant (SV) analysis in 50 dilated cardiomyopathy patients, and associates these variants with gene expression. The authors conclude significant levels of contribution of SVs to gene expression variability between subjects.

Overall, despite recent publications reporting on SVs in cancer or in general population in relation to gene expression, and despite the largely descriptive nature of the manuscript, this is an informative study, particularly for DCM. The study was well designed and executed.

My main concern is that, despite the authors being very careful to examine SV-eQTL pairs within LD intervals, it is still difficult to attribute the causality of the eQTLs to be the corresponding SVs, as the authors cannot exclude the contribution of other variants, such as linked SNPs, as the cause of the eQTLs. This is of course not easy to address. The authors may do a thorough SNP analysis on this same dataset, but that will probably take some time and probably cannot dissect the causality either. I suggest the authors to do four things.

1. Pay attention to the choice of words throughout the manuscript, and change those that indicate or imply causality, unless there is substantial evidence.
2. For SVs that overlap with gene coding regions, it is highly likely that the SVs are the cause of the eQTLs. The authors went into length on specific examples, but did not give a summary of the numbers of SV-eQTLs in which the SV cover coding part, intron part, and other neighboring regions of the eQTL gene(s). I suggest the authors to group the SV-eQTLs into several categories reflecting SV's location in affected eQTL genes, as well as considering directionality of the change-for example, if the SV is a deletion but eQTL gene is going up with the SV presence, it is more difficult

to argue for a direct causal relationship.

3. It will also be helpful to show RNAseq traces (may have to be averaged among subjects) for specific examples of SV-eQTLs that affect exonic regions. For example, in Figure 5b, so readers can appreciate how RNA levels are affected throughout this transcript. If there are specific dips at the deleted exons, it will be much more convincing that these are causal.

4. For the trans effect analysis in which the authors identify ~7.5% or >10% of the gene expression variation to be explained by the SVs, it can be very misleading given this main concern. The authors should discuss major caveats with these statistics.

Minor:

1. Figure 2: It is unclear how enhancer, TFBS etc were defined or obtained. Were they defined based on enhancer and TFBS analysis in the heart tissues or related cell types? Similarly, it is unclear how lincRNA regions were defined or obtained. The authors have the opportunity to use their own RNAseq data to define lincRNAs in left ventricle. It is also unclear what protein-coding region means—please clarify if these are introns, exons, or fractions of open reading frames.

2. Figure 2: this seems to be an analysis based on the union of all SVs. Is it possible that the authors provide a distribution among patients on these categories, so that the readers can appreciate the potential impact in individual patients?

3. Figure 4: it will be helpful to label the number of subjects for each variant type in the bar plots—for example, the homo KRT1 SV seems to be only from a single patient?

4. Figure 5 and sup Fig 3: it will be helpful to show the genomic structure of the large locus (with the neighboring genes), rather than simply for a single gene.

1st Revision - authors' response

11 August 2017

Referee #1

This is a detailed study that is the first study to demonstrate genomic variability due to SVs in DCM. It is competently performed study that provides information on a large number of SVs present in DCM patients. However, it is a descriptive study, and the authors over-conclude what the implications of these SVs have on DCM disease pathology. It is unfortunate that the authors did not include a control group of patients without DCM to determine which of the SCVs are specific for the patients with DCM. A comparison of population matched controls was performed by in silico re-genotyping of all identified SVs from the 1000 Genomes Project, but this is not the same as including their own control patients.

We thank the reviewer for acknowledging the novelty and high quality of the implementation of our study. The association analysis for the impact of SVs on the DCM phenotype were now removed due to the advice of the editor. We hence focused on the strong association statistics for the effect of SVs on gene expression in the failing heart. To supplement this strategy, we included SNV-eQTL analysis as requested by the reviewers and compared the beta-values of both approaches. As much as 65.3% of the SV-eQTLs do not harbour a significant association with a single nucleotide variant. For the overlapping SV/SNV-eQTL the directional beta values of an additive gene effect model are comparable (Figure 3). For directly deleted protein-coding exons, we find a strong correlation with reduced expression (Figure 2d/e), for the low number of duplications we did only find a modest trend to increased expression. For more distant SVs, we included Western-Blot experiments for SERPINC1, showing that a heterozygous carrier status is linked to increased mRNA and protein expression.

This in silico analysis revealed 538 novel deletions and 194 novel duplications in human DCM patients. Comparison of the SVs obtained in the DCM patients with the subjects of the 1000 Genomes Project identified a number of novel genes that are proposed to be involved in the pathogenesis of DCM, including ADCYAP1, ARHGAP1, DAXX, SERPINC1, EPX, CLIC2, and CAMK2A. While the authors discuss the potential link of these changes to DCM pathology, the actual link between alterations in these genes to DCM pathology is weak. It is also unfortunate that no data is provided as to whether the actual expression of the proteins associated with these genes is altered in DCM. This, of course, is complicated by the lack of a control subject group in this study.

We totally agree that our descriptive study design is not sufficient to draw conclusions on the causal involvement of SVs or SV-eQTLs on the pathogenesis of DCM. We believe that our study is pioneering the research on SV-eQTLs in patients and wanted to provide an outlook on the potential of this class of under-investigated genetic variation. To follow your valuable advice and that of the editor, we reworded our manuscript to be very sensitive for this matter and avoided over-interpretation.

Since we believe that readers of different fields of molecular research should be equally addressed, we incorporated explorative data from transverse aortic constriction mice for the SV-eQTLs with homologous gene in mice, showing that *Gstt1/2* (**Figure 5d**) and *Gabarapl* (**Suppl. Fig. 5**) are significantly altered during the pathogenesis of heart failure (adaptive or maladaptive).

The link between SV and gene-expression is supported by the association statistics visualized by Manhattan plots. We now included exemplary protein analysis of *SERPINC1* in SV-mutation carriers and wildtypes, underlining that cis-regulatory SVs outside of coding exons do impose a significant alteration on protein levels.

Together, we provide a unique insight into the role of SVs on the cardiac transcriptome of a considerable number of patients and provide the community with estimates on their number, effect sizes and potential to identify novel targets for heart failure research.

It is not clear whether any of the proposed SVs seen in the genome result in adaptive or maladaptive changes in DCM. Therefore, the authors need to be cautious in linking these SVs to DCM pathology.

We agree and revised our wording throughout the manuscript. From the now included TAC data, we see at least for 3 gene products that they are significantly dysregulated during this hypertrophy/failure response. We carefully introduced these findings and avoided overinterpretation (page 9).

Specific Comments:

1) Abstract, last paragraph: The authors suggest that they pinpoint the genomic variability due to SVs and demonstrate their specific molecular effect. This is an overstatement and no specific molecular effects are demonstrated.

We agree, we provide an association analysis providing a link between genetic variability and cardiac gene expression. We hence revised our wording.

2) Introduction: para 1, line 3: The references cited do not discuss the cascade involving cardiac energy metabolism implied in the sentence.

We thank the reviewer for pointing out the missing reference for energy metabolism in the heart. We therefore added the references of Tuomainen & Tavi, and Mizushima *et al.*, which describe how the variation of gene expression and DNA binding proteins are involved in the metabolism of healthy heart and in the progression of heart diseases (Mizushima, Takahashi *et al.*, 2016, Tuomainen & Tavi, 2017). We also included references for energy metabolism during heart failure and related genetic variation in Black 6 (C57BL/6J) mice (Nickel, von Hardenberg *et al.*, 2015).

3) Introduction, para. 2, line 1: The authors suggest that changes in gene expression are "adaptive". The possibility that they may be maladaptive is not discussed.

We apologize for neglecting the maladaptive consequences through adaptive transcriptional mechanisms and updated the sentence accordingly, which now reads: "*Diverse mechanisms are known to contribute to the adaptive and maladaptive gene expression in the human myocardium, such as transcription factors and its binding sites, micro- and circular-RNAs, lncRNAs, histone modifications and direct chemical changes of the DNA (Chang & Han, 2016, Haas, Frese *et al.*, 2013, Lighthouse & Small, 2016)*"

Referee #2

This manuscript reports structural variant (SV) analysis in 50 dilated cardiomyopathy patients, and associates these variants with gene expression. The authors conclude significant levels of contribution of SVs to gene expression variability between subjects.

Overall, despite recent publications reporting on SVs in cancer or in general population in relation to gene expression, and despite the largely descriptive nature of the manuscript, this is an informative study, particularly for DCM. The study was well designed and executed.

My main concern is that, despite the authors being very careful to examine SV-eQTL pairs within LD intervals, it is still difficult to attribute the causality of the eQTLs to be the corresponding SVs, as the authors cannot exclude the contribution of other variants, such as linked SNPs, as the cause of the eQTLs. This is of course not easy to address. The authors may do a thorough SNP analysis on this same dataset, but that will probably take some time and probably cannot dissect the causality either.

We thank the reviewer for appreciating our strategy to identify SV-eQTLs in our DCM cohort. As we totally agree that it is important to also account for SNVs, we have now also performed an SNV-eQTL. For 49 of the 75 SV-eQTLs (65.3%) we did not find any SNV to account for an expression change as predicted by the SV-eQTL. In case of 25 SV-eQTLs, we cannot rule out an additive effect mediated by SV-eQTLs and SNV-eQTLs. We have also added a statement from a recent meta-study analysing SVs across tissues, who “*estimated that 3.5–6.8% of all cis-eQTLs (SNV or SV) are driven by a causal SV*” (PMID: 28369037).

I suggest the authors to do four things:

1. Pay attention to the choice of words throughout the manuscript, and change those that indicate or imply causality, unless there is substantial evidence.

As proposed, we modified the wording of our statements throughout the manuscript. We e.g. updated the last paragraph of the discussion to improve the understanding of the presented results and avoid misinterpretation, which now reads: “*This immediately raises the question about their pathophysiological role in DCM, which cannot be answered by performing association analyses, only. Subsequent studies on the functional role of the identified targets need to be performed in line with large-scale studies including an excellently phenotyped control cohort matched for population ancestry, age and gender.*”

In addition, we also removed the association analysis with the 1000genomes control cohort to avoid over-interpretation of the calculated associations. This was a suggestion of the Editorial Board.

2. For SVs that overlap with gene coding regions, it is highly likely that the SVs are the cause of the eQTLs. The authors went into length on specific examples, but did not give a summary of the numbers of SV-eQTLs in which the SV cover coding part, intron part, and other neighboring regions of the eQTL gene(s). I suggest the authors to group the SV-eQTLs into several categories reflecting SV's location in affected eQTL genes, as well as considering directionality of the change-for example, if the SV is a deletion but eQTL gene is going up with the SV presence, it is more difficult to argue for a direct causal relationship.

We thank the reviewer for the constructive comment. We now restructured **Figure 2** to now include information on the number of affected exons and introns alike. We further added **Figure 4a** which gives detailed information on the up-/down- regulation of the genes which are linked to the different genomic loci / functional elements (TFBS, enhancer, etc.). Together with recent evidence from other studies on SVs, we believe that this complex cis-regulatory effect can well be explained by the altering effects on regulatory elements (Chiang et al., 2017). As an example, we also performed Western Blot analysis on cardiac protein expression of one of the SV-eQTLs (SERPINC1), which underlines the profound effect of cis-SVs on protein coding genes.

3. It will also be helpful to show RNAseq traces (may have to be averaged among subjects) for specific examples of SV-eQTLs that affect exonic regions. For example, in Figure 5b, so readers can appreciate how RNA levels are affected throughout this transcript. If there are specific dips at the deleted exons, it will be much more convincing that these are causal.

It can be difficult to visually interpret the RNA seq traces, since often larger regions of several kilobases are affected and the beta values of expression changes are relatively low (in the same range as for SNV-eQTLs, see figure 3b). To follow your advice we exemplarily show the effect of an exonic deletion (new figure 2d) and the associated RNA-seq traces. Here, the heterozygous carriers have only 59% of the expression of the wild-type patients. Importantly, homozygous carriers (n=3) do not show remaining expression in the region of the deletion. Close to the SV-event there is still a substantial effect on mRNA expression (right side of the deletion, blue bar).

4. For the trans effect analysis in which the authors identify ~7.5% or >10% of the gene expression variation to be explained by the SVs, it can be very misleading given this main concern. The authors should discuss major caveats with these statistics.

We have updated the methods section which now reads: *“From each SV-eQTL with P-value of 1% or less, a residual variance was computed. For a set of SV linked genes, the residual variance of each gene was summed to generate a residual variance of the set of genes. Together with the total variance of the set, the coefficient of determination of the set of genes was then calculated accordingly, i.e. $1 - \sum(\text{residual variances})/(\text{total variance})$. The value represents then the proportion of total variance that the model explains. The coefficients of determination are meant to be descriptive, hence no associated statistical significances are calculated.”*

In the discussion we modified the description to now read: *“The described findings underline the high potential of SVs to act on cardiac gene expression in a multifaceted manner (Zhang & Lupski, 2015). Chiang et. al. recently estimated that 3.5–6.8% of all cis-eQTLs are driven by a SV (Chiang, Scott et al., 2017). This immediately raises the question about their pathophysiological role in DCM. We performed stringent statistics and focused on cis regulatory effects within linkage disequilibrium to provide robust estimates on the relevance of SVs in this context. When including trans regulation, as much as 7.5% of the whole myocardial expression variation could be explained by SVs and 10.1% of the variation of DCM-related genes (KEGG). Besides this statistical evidence of association, we however cannot proof a causal relationship with our study design. Subsequent investigations on the functional role of the identified targets need to be performed in line with large-scale studies including excellently phenotyped control cohorts.”*

Minor:

1. Figure 2: It is unclear how enhancer, TFBS etc were defined or obtained. Were they defined based on enhancer and TFBS analysis in the heart tissues or related cell types? Similarly, it is unclear how lincRNA regions were defined or obtained. The authors have the opportunity to use their own RNAseq data to define lincRNAs in left ventricle. It is also unclear what protein-coding region means-please clarify if these are introns, exons, or fractions of open reading frames.

We thank the reviewer for pointing this out and now used our RNA expression data from the human heart to identify the most abundant transcript for each gene. We have described this approach in the figure legend of figure 4a, which now reads: *“Shown is the number of regulatory genomic regions hit by SV-eQTLs leading to a positive (grey) or negative (black) gene regulation. Annotation is based the most abundant transcript per gene based on the RNA-seq data.”*

Additionally, besides the already cited TFBS database, we included the reference for the enhancers (page 23).

2. Figure 2: this seems to be an analysis based on the union of all SVs. Is it possible that the authors provide a distribution among patients on these categories, so that the readers can appreciate the potential impact in individual patients?

This is an excellent idea. We performed the analysis and introduced them into figure 2c (bottom, spider plots). As shown, there are suspicious patients that carry a multiple of affected regulatory sites that are deleted (or duplicated). We more closely investigated those patients and identified that they carry large genetic deletions of up to 10 mega bases.

3. Figure 4: it will be helpful to label the number of subjects for each variant type in the bar plots-for example, the homo KRT1 SV seems to be only from a single patient?

We have now added the number of heterozygous and homozygous variant carriers and non-affected DCM patients to the respective boxplots (Figure 4 and 5 and Supplemental Figure 5 and 6). For KRT1 there was indeed only 1 homozygous carrier.

4. Figure 5 and sup Fig 3: it will be helpful to show the genomic structure of the large locus (with the neighbouring genes), rather than simply for a single gene.

We follow your advice and visualized the genomic structure of new Figure 5 and Supplemental Figure 4 (old suppl. Fig. 3).

References:

- Chiang C, Scott AJ, Davis JR, Tsang EK, Li X, Kim Y, Hadzic T, Damani FN, Ganel L, Consortium GT, Montgomery SB, Battle A, Conrad DF, Hall IM (2017) The impact of structural variation on human gene expression. *Nat Genet* 49: 692-699
- Mizushima W, Takahashi H, Watanabe M, Kinugawa S, Matsushima S, Takada S, Yokota T, Furihata T, Matsumoto J, Tsuda M, Chiba I, Nagashima S, Yanagi S, Matsumoto M, Nakayama KI, Tsutsui H, Hatakeyama S (2016) The novel heart-specific RING finger protein 207 is involved in energy metabolism in cardiomyocytes. *J Mol Cell Cardiol* 100: 43-53
- Nickel AG, von Hardenberg A, Hohl M, Löffler JR, Kohlhaas M, Becker J, Reil JC, Kazakov A, Bonnekoh J, Stadelmaier M, Puhl SL, Wagner M, Bogeski I, Cortassa S, Kappl R, Pasięka B, Lafontaine M, Lancaster CR, Blacker TS, Hall AR et al. (2015) Reversal of Mitochondrial Transhydrogenase Causes Oxidative Stress in Heart Failure. *Cell Metab* 22: 472-84
- Tuomainen T, Tavi P (2017) The role of cardiac energy metabolism in cardiac hypertrophy and failure. *Exp Cell Res*
- Zhao J, Li D, Seo J, Allen AS, Gordan R (2017) Quantifying the Impact of Non-coding Variants on Transcription Factor-DNA Binding. *Res Comput Mol Biol* 10229: 336-352

2nd Editorial Decision

26 September 2017

Thank you for the submission of your revised manuscript to EMBO Molecular Medicine. We have now received the enclosed reports from the reviewers that were asked to re-assess it. As you will see the reviewers are now supportive, although #2 has a few final requests that require your action.

I am prepared to make a quick editorial decision on your revised manuscript provided you carefully deal with the reviewer's final requests and fulfil the following editorial requirements:

- 1) The accession number(s) for the deposited data must be made available in the manuscript in a "Data Availability" section of the Material and Methods. You may refer to our guidelines on data deposition. Please note that we cannot proceed with acceptance without this information.

Please submit your revised manuscript within two weeks. I look forward to seeing a revised form of your manuscript as soon as possible.

***** Reviewer's comments *****

Referee #1 (Comments on Novelty/Model System for Author):

In my original review I stated that the study would be stronger with the inclusion of "control" data. This was not done, but I recognize the difficulty of doing this. The authors have, however, tempered their conclusions and interpretation of the data based on this lack of control data.

Referee #2 (Comments on Novelty/Model System for Author):

This revision has addressed almost all concerns raised previously.

Referee #2 (Remarks for Author):

The revised manuscript has addressed all my main concerns.

There are a few minor issues:

1. In abstract "duplication events, however, do show ambiguous effects." The word ambiguous is probably not the best word, which makes this sentence difficult to understand without reading the manuscript. How about "do not show significant changes as a group"?
2. In abstract "In summary, we are first to describe the genomic variability due to SVs in heart failure due to DCM and dissect their impact on the transcriptome". With the concerns on the lack of causality, I suggest the authors to use "... genomic variability associated with SVs".
3. Page 6: "In the to our knowledge most detailed...." Seems to be a typo?
4. Page 10: "we inferred the degree of global SV-mediated expressional variation to be ...". Please change "SV-mediated" to "SV-associated"

2nd Revision - authors' response

29 September 2017

In detail, we revised the following minor points:

1.) In abstract "duplication events, however, do show ambiguous effects." The word ambiguous is probably not the best word, which makes this sentence difficult to understand without reading the manuscript. How about "do not show significant changes as a group"?

We have changed the wording, the sentence now reads: "*In cases of deleted protein coding exons, we find downregulation of the associated transcripts, duplication events, however, do not show significant changes as a group*".

2.) In abstract "In summary, we are first to describe the genomic variability due to SVs in heart failure due to DCM and dissect their impact on the transcriptome". With the concerns on the lack of causality, I suggest the authors to use "... genomic variability associated with SVs".

We have changed the wording accordingly.

3.) Page 6: "In the to our knowledge most detailed...." Seems to be a typo?

We revised the sentence accordingly, which now reads: "*To our knowledge, we present the most detailed map of genetic variation in a well phenotyped cardiomyopathy cohort and find that the genome-wide SVs delete important loci directly related to gene expression, such as enhancers (n=875), TFBSs (66,147) and lncRNAs (3,100) (Fig 2C)*".

4.) Page 10: "we inferred the degree of global SV-mediated expressional variation to be ...". Please change "SV-mediated" to "SV-associated"

We have changed the wording accordingly.

Corresponding Author Name: Dr. Benjamin Meder

Manuscript Number: EMM-2017-07838